# Analysis of lncRNAs in *Lupinus mutabilis* (Tarwi) and Their Potential Role in Drought Response

**DOI:** 10.3390/ncrna9050048

**Published:** 2023-08-23

**Authors:** Manuel Hidalgo, Cynthia Ramos, Gaston Zolla

**Affiliations:** 1Programa de Estudio de Medicina Humana, Universidad Privada Antenor Orrego, Av. América Sur 3145, Trujillo 13008, Peru; jemhidalgor@gmail.com (M.H.); cynthiaramosotiniano@gmail.com (C.R.); 2Laboratorio de Fisiología Molecular de Plantas del Programa de Cereales y Granos Nativos, Facultad de Agronomía, Universidad Nacional Agraria La Molina, Lima 12, Peru

**Keywords:** legume, Andean lupin, lncRNA, drought, SECIS-like element

## Abstract

*Lupinus mutabilis* is a legume with high agronomic potential and available transcriptomic data for which lncRNAs have not been studied. Therefore, our objective was to identify, characterize, and validate the drought-responsive lncRNAs in *L. mutabilis*. To achieve this, we used a multilevel approach based on lncRNA prediction, annotation, subcellular location, thermodynamic characterization, structural conservation, and validation. Thus, 590 lncRNAs were identified by at least two algorithms of lncRNA identification. Annotation with the PLncDB database showed 571 lncRNAs unique to tarwi and 19 lncRNAs with homology in 28 botanical families including Solanaceae (19), Fabaceae (17), Brassicaceae (17), Rutaceae (17), Rosaceae (16), and Malvaceae (16), among others. In total, 12 lncRNAs had homology in more than 40 species. A total of 67% of lncRNAs were located in the cytoplasm and 33% in exosomes. Thermodynamic characterization of S03 showed a stable secondary structure with −105.67 kcal/mol. This structure included three regions, with a multibranch loop containing a hairpin with a SECIS-like element. Evaluation of the structural conservation by CROSSalign revealed partial similarities between *L. mutabilis* (S03) and *S. lycopersicum* (Solyc04r022210.1). RT-PCR validation demonstrated that S03 was upregulated in a drought-tolerant accession of *L. mutabilis*. Finally, these results highlighted the importance of lncRNAs in tarwi improvement under drought conditions.

## 1. Introduction

In recent years, the study of plant genomes has shown that non-coding sequences are essential components of the genetic machinery that perform different functions [1]. Non-coding sequences can be divided into two groups: the first group includes the small RNAs (microRNAs, small nuclear RNA, small nucleolar RNA, tRNA-derived small RNA, and Piwi-interacting) [2,3,4]; the second group includes the long non-coding RNAs (lncRNA). By definition, the lncRNAs are sequences longer than 200 bp that do not code for proteins. However, they can still regulate gene expression at different levels, including chromatin remodeling, transcription, post-transcriptional processes, and translation [5].

In plants, these lncRNAs are differentially expressed and regulate differentiation, maturation, reproduction, gene silencing, and environmental response, including stress factors [6,7,8]. lncRNAs have been discovered in different plant species, where they perform essential functions in stress responses. For example, the drought-induced lncRNA (DRIR) is transcribed in *Arabidopsis thaliana* in response to drought and salt stress [9]. Other lncRNAs can regulate flowering time in the same species by controlling flowering locus C (FLC) expression through epigenomic changes [10]. In *Manihot esculenta*, Li et al. [11] found 318 lncRNAs, with 16 lncRNAs acting as miRNA mimetic targets in response to drought and cold. In *Pyrus betifolia*, 251 stress-responsive lncRNAs were related to desiccation [12]. Kim et al. [13] found differentially expressed lncRNAs involved in transcription factor regulatory networks that regulated growth in *Zea mays*.

lncRNAs have also been identified in legumes, the third biggest family of flowering plants, which have low cost and high nutritional value and are essential components of the human diet [14]. Some species studied are *Cicer arietinum*, which were studied by Khemka et al. [15], who found 2248 lincRNAs involved in growth during eight floral and three vegetative stages of development. In *Glycine max* and *G. soja*, Lin et al. [16] found 69,000 loci for RNA genes after sequencing 332 samples with specific transcriptional responses for different tissues and phenological stages. In other legumes, the genetic control of the flowering process has been studied by Hidalgo et al. [17], demonstrating that the lncRNA EL0144 is involved in floral development and abiotic responses in *Lupinus angustifolius*. Likewise, Aslam et al. [18] analyzed lncRNAs in *L. albus* grown under phosphorous deficiency and found 1564 intergenic lncRNAs and 464 natural antisense intergenic transcripts (NAT) that participate in the regulation network under P-deficiency stress. Similarly, different lncRNAs regulate drought response by forming potential subtle regulatory networks that control multiple genes to determine the overall response of plants [19]. Some lncRNAs are miRNA precursors that interact with stress-responsive elements such as transcription factors at a molecular level [4], demonstrating their importance in drought stress.

Drought threatens food security by limiting crop production, which causes drastic losses of potential yield. This affects worldwide populations, mainly those living in arid and semi-arid areas [20], where droughts have become significantly more frequent. In the Peruvian highlands, during 2022, the lack of rainfall caused the most severe drought in 50 years [21]. Furthermore, drought causes health problems, such as malnutrition, due to the decreased availability of food, including micronutrient deficiency [22]. Thus, it is essential to understand the drought response in superfoods to withstand drought stress and reduce the risks of malnutrition.

Tarwi (*L. mutabilis* Sweet) is an ancient legume crop from the Andes highlands, which has recently been regarded as a superfood for its remarkably high nutritional value [23]. Moreover, it is biofortified for Fe, Zn, and B [24] and has bioactive alkaloids with antimicrobial properties [25]. Additionally, the availability of tarwi RNA-seq data [26,27] enables lncRNA identification. It provides an opportunity to understand the evolutionary landscape of lncRNAs, conservation, and functionality. Therefore, this study aims to identify, characterize, and validate the lncRNAs in *L. mutabilis* under drought stress.

## 2. Results and Discussion

### 2.1. lncRNA Identification

The transcriptomic data included 222,217 transcripts with a 200 bp length (Figure 1A). Then, 120,493 transcripts with ORFs longer than 300 bp that might include coding regions were removed [28]. The remaining 101,724 transcripts were compared against the Pfam and SwissProt databases; thus, 100,300 transcripts were discarded as they included coding domains. Following the methodology of Khemka et al. [15], in *L. mutabilis*, we found 1424 putative non-coding sequences as opposite to the 3051 ones reported in *C. arietinum*.

To improve the detection of lncRNAs, we used different algorithms (Appendix A) that evaluate the coding potential [29] (Figure 1B). The algorithm CPC [30] has been evaluated and performs well in eukaryotic lncRNA identification [31]. The *L. mutabilis* data allowed the identification of 625 lncRNAs. In contrast, Aslam et al. [18] found 2028 lncRNAs in *L. albus* roots under P deficit and Khemka et al. [15] found 2248 lincRNAs in *C. arietinum* using the same algorithm. The higher number count for lncRNA in these studies, in contrast to the 625 *L. mutabilis*, might be due to the inclusion of more developmental stages.

The CPC algorithm is based on a vectorial support machine (SVM) that includes six characteristics in their predictive model, including the coverage of the ORFs and sequence similarities to protein-coding genes [30]. However, developing more precise algorithms enables better characterization of lncRNAs [29]. These algorithms used in combination can improve the lncRNA prediction [18,32]. Hence, the algorithms included in this research were CPC, CNIT, CPC2, and lncFinder (Figure 1B).

The CNIT algorithm enabled the identification of 266 lncRNAs; 243 lncRNAs were shared with CPC2, 255 lncRNAs were common with lncFinder, and 1 lncRNA was common with CNIT (Figure 1B). The CNIT algorithm is based on CNCI and performs the lncRNA identification of different eukaryotic organisms, including *A. thaliana* [33]. This algorithm has been used by Wang and Nambeesan [34] to find 919 and 1116 lncRNAs in seven fruit stages in two species of *Vaccinium*. On the other hand, Glushkevich et al. [35] used the CNIT algorithm to identify 4382 lncRNAs in *Solanum tuberosum*.

The use of CPC2 enabled the identification of 429 putative lncRNAs in *L. mutabilis*. A total of 243 lncRNAs were shared with CNIT, 387 lncRNAs were common to the lncFinder results, and 32 lncRNAs were unique to CPC2. The CPC2 algorithm predicts the lncRNAs by analyzing the ORF length, integrity, the Fickett score, and the isoelectric point [30] and has been used by several authors to analyze plant transcriptomes. For example, Sang et al. [36] analyzed 25 plant species and found 8817 lncRNAs in *G. max* and 10,904 lncRNAs in *M. truncatula*. Likewise, Ma et al. [37] identified four lncRNAs involved in anthocyanin biosynthesis in *Malus domestica* fruits. At the same time, Palos et al. [38] used CPC2 to reannotate 2657 lncRNAs in the Brassicaceae family.

Lastly, with lncFinder, 547 putative lncRNAs were identified. These 255 lncRNAs were shared with CNIT and 387 with CPC2. The lncFinder algorithm evaluates the heterologous characteristics of the sequence using a machine-learning approach [39] and has been used in other plant species for lncRNA prediction. In *Mangifera indica*, Moh et al. [40] used lncFinder values equal to or less than 0.5 to identify lncRNAs involved in growth, developmental processes, defense against pathogens, and stress response. lncFinder was also used in *Hevea brasiliensis* by Wang et al. [41] to characterize 12,029 lncRNAs. In total, 233 lncRNAs were common to all four algorithms used in the lncRNA identification of *L. mutabilis* (Figure 1B); the distribution of transcripts according to coding probability for each algorithm is shown in Figure 1C.

The coincidence between two or more applied algorithms was taken to improve lncRNA prediction and further characterize putative lncRNAs, following Aslam et al.’s [18] criterion in *L. albus*. Hence, 590 transcripts were taken as putative lncRNAs in *L. mutabilis*. Characterizing these transcripts is important because some sequences regarded as lncRNAs can still produce small peptides, as reported by Xing et al. [42]. Therefore, Duan et al. [31] recommended a functional analysis, including the annotation, thermodynamic characterization, and validation of lncRNAs.

### 2.2. Annotation of Putative lncRNAs

Direct annotation of lncRNAs has been challenging due to the absence of genomic data, partial annotation of protein-coding genes, and limited tools to assemble transcripts from short reads [43]. The annotation followed the reconstruction and filtering of the 590 transcripts. The comparison of the *L. mutabilis* performed the annotation lncRNAs with all the transcripts present in the PLncDB v2.2 database, following the homology criterion of Hezroni et al. [43], complemented by Khoei et al. [44]. Figure 2A shows an upset plot including the groups generated in the 28 botanical families represented in the PLncDB v2.2 database. We found 571 unique transcripts for *L. mutabilis*; 19 transcripts showed homology with different families. The families with the highest number of homologs with *L. mutabilis* lncRNAs were Solanaceae (19), Fabaceae (17), Brassicaceae (17), Rutaceae (17), Rosaceae (16), and Malvaceae (16), among others. Furthermore, 14 lncRNAs were common to more than 10 families. The low number of conserved sequences found in the different families demonstrates the poor conservation of lncRNAs and the high evolutionary rate of these sequences. These results coincide with the findings of Sang et al. [36], who found 756 conserved lncRNAs in *A. thaliana.*

The systematic annotation of orthologous lncRNAs indicates the existence of biological functions maintained throughout the evolutionary process [45,46]. The conservation across lncRNAs might indicate important functions in controlling different cellular processes, so lncRNAs with sequence conservation are particularly interesting to study [43]. Therefore, we selected the transcripts common to at least 40 plant species. The species with 15 or more lncRNAs homologs to tarwi included *Capsicum annuum* (19), *Citrus sinensis* (16), *G. max* (16), *Gossypium barbadense* (16), *Arachis ipaensis* (15), *C. arietinum* (15), *Elaeis guineensis* (15), *G. raimondii* (15), *M. truncatula* (15), *Nicotiana tabacum* (15), *S. lycopersicum* (15), *S. tuberosum* (15), *Trifolium pratense* (15), and *Vigna radiata* (15), which indicates the low conservation in lncRNA sequences. The lower conservational pattern is present in A. thaliana, a species with few conserved RNAs according to Liu et al. [47]. In contrast, the lncRNA conservation between sorghum and maize reached up to 25% of the sequences according to Li et al. [48]. A total of 12 lncRNAs showed homology to 40 species or more (Figure 2B). These transcripts were further taken for the subcellular location and thermodynamic characterization.

### 2.3. Cellular and Subcellular Location

The location of each *L. mutabilis* lncRNA was assessed with four tools, including RNAlocate [49], iLoc-lncRNA [50], DeepLncLoc [51], and lncLocator [52] (Table 1). The results indicated that 67% of the transcripts were in the cytoplasm, with 25% lncRNAs located in the cytosol, where they might have different roles in gene regulation and expression by interacting with mRNAs, ribosomes, and proteins [53]; 75% acted at the level of ribosomes, participating in transcription [54]. On the other hand, 33% of sequences were located in the exosomes; they were composed of proteins and RNA that participate in RNA metabolism by silencing repetitive sequences through degradation [55]. The exosomes contained lncRNAs that might modulate the interaction among the regulatory elements that control the genetic expression and the nuclear organization by regulating other lncRNAs produced from these elements [56].

### 2.4. lncRNA Thermodynamic and Structural Characterization

The polynucleotide chain of an RNA molecule constitutes its primary structure; it folds as a result of base-pairing, generating higher-order structures, such as secondary and tertiary ones [57]. The folding process is hierarchical, depending on the base pairing that shapes the secondary structure and accounts for most of the thermodynamic energy of each structure [58]. Also, from a thermodynamic perspective, the most stable structure appears when the free energy is minimized [59] and is used along with other criteria for identifying functional lncRNAs [60]. In this research, we used four algorithms, including RNAfold [61], LinearFold [62], LinearPartition [63], and RNAshapes [64], to measure the minimum free energy (MFE) of each lncRNA (Figure 3A).

The sequences S02 and S12 showed values over −10 kcal/mol. Nine transcripts (S01, S04, S05, S06, S07, S08, S09, S10, and S11) showed MFE values between −10 and −50 kcal/mol. Meanwhile, transcript S03, with homology in 41 species, had an average MFE value of −105.67 kcal/mol (Figure 3A), which can be considered stable [65]. The stable higher-order structures were more likely to contain high-information-content clefts and pockets capable of interacting with other molecular elements [66]. Therefore, due to its low MFE value, S03 might form structures of higher order involved in *L. mutabilis* genetic regulation.

Since the tertiary structure of RNA is highly limited by the bi-dimensional structure [67], we analyzed the secondary structure of S03 generated by RNAFold Delta (Figure 3B). An RNA molecule contains rigid and ecstatic structures and free highly dynamic ones [68], which can be organized into regions. According to Ding et al. [69], these RNA regions contain distinctive structural motifs grouped into molecules. The structure of the lncRNA S03 showed three distinctive and well-defined regions, including two multibranch loops and a short stem with an internal loop (Figure 3B), described according to Bugnon et al. [70].

Each region had several motifs (loops, stems, hairpins, and bulges, among others), which are tridimensional structural patterns that emerge from a particular group of interactions that provide stability and functionality [71]. Region I showed a multibranch loop (Figure 3(BI)) that included six hairpins with terminal loops (h1, h2, h3, h4, h5, and h6) and two short stems (s1 and s2). The hairpins h1 and h2 had two internal loops, while h3, h4, and h5 had a big terminal loop and h6 had two bulges with a terminal loop in its stem. The complexity observed in Region I indicated the potential presence of structures with specific functions interacting with other elements [72].

Region II (Figure 3(BII)) included a short stem with an internal loop. According to Ross and Ulitsky [72], this type of region tends to be less important at a functional level because it is simple, does not include other elements, and is more flexible. This flexibility allows them to acquire different conformations at a tridimensional level by reorganizing the secondary structures to form new tertiary structures [73]. The conformations produced can vary in space since they depend on interactions that might occur at physiological temperatures, producing a change in the overall MFE of the molecule [74]. These properties enable the connection of functional elements [68], as seen in S03, in which region II connected the more complex regions I and III. This flexibility might indicate that the configuration of this RNA molecule, as described by Mustoe et al. [75], is not invariable in time. Thus, the RNA molecules can acquire different tertiary structures at a rate that depends on the energetic barriers that separate the different conformations [74], as might be the case for S03.

Region III (Figure 3(BIII)) was connected to an internal loop of region II and also had a complex structure, including a second multibranched loop with two short stems (s3 and s4) and four hairpins with bulges and internal loops (h7, h8, h9, and h10). The hairpins h7, h9, and h10 had 2–4 bulges and h9 and h10 included internal loops. Likewise, h7, h8, h9, and h10 had an external loop. These structures can allow for the interaction of lncRNAs with proteins that regulate transcription and translation [76]. In particular, the hairpin h10 had a structure similar to the selenocysteine insertion sequence (SECIS-like element), including a stem–loop structure with an AGU sequence next to the second bulge.

Structures such as stems with bulges and internal loops, as observed in regions I and III of S03, including h1, h2, h7, h9, and h10 (Figure 3B), might indicate that this transcript has biological functionality. Indeed, these structures have been involved in the interaction between lncRNAs and proteins and can regulate transcription, splicing, and translation [76]. Moreover, according to Li et al. [77], the loops, bulges, and hairpins found in these structures might be involved in the interaction with mRNA and the regulation of different metabolic processes, as proposed with the SECIS-like element in this study.

According to Svoboda and Di Cara [78], the SECIS element is a cis-acting structural element that lies next to the UGA codon and allows it to synthesize de novo selenocysteine. This selenoamino acid can be uptaken by membrane transporters in the roots [79]. The SECIS element is a binding site for SECIS-binding protein 2 (SBP2), redefining the UGA codon from STOP to the amino acid selenocysteine in a growing peptide [80]. This mechanism is part of the RNA processing of selenoproteins and can be extended to non-coding sequences, as was demonstrated by Mita et al. [81], who found SECIS-like elements in the lncRNA CCDC152. This lncRNA can reduce the expression of selenoprotein P by interacting with the mRNA, which inhibits the binding to the SECIS-binding protein 2 and reduces its affinity for the ribosomes, regulating gene expression [81].

Fajardo et al. [82] reported the SECIS element in vascular plants, providing insights into the genetic mechanisms that regulate Se metabolism in plants. Indeed, Frias et al. [83] found selenometilselenocystein in *L. angustifolius* seeds. The role of selenoproteins might be undermined by the need for more comprehension of the molecular mechanisms that control different processes at a genetic level. One of these proteins is molybdenum cofactor sulfurase (LOS5/ABA3), expressed in tarwi under drought stress [27]. LOS5/ABA3 is a selenocysteine lyase that generates a molybdenum cofactor [84] necessary for the aldehyde oxidase activity in a plant’s final step of ABA biosynthesis [85]. Thus, LOS5/ABA3 might influence the stress response in plants.

Furthermore, ABA and selenium have been reported to alleviate drought stress in several plants [86]. Remarkably, selenoproteins O is associated with ROS scavenging and dehydration tolerance in *A. thaliana* [87] and is involved in the regulation of REDOX functions by inactivating H_2_O_2_ and other toxic compounds [87]. However, more research is necessary to understand the Se homeostasis under drought and the regulation of its metabolism by lncRNAs.

### 2.5. Structural Conservation

Functional lncRNAs have relatively low sequence identity but secondary structures conserved across thousands of sequences [88]; these structures depend on the pairings of multiple nitrogen bases that generate stable secondary structures that cannot be disrupted at physiological temperatures. Thus, the structural conservation of the lncRNA S03 with its 41 homologs (Appendix A) was evaluated by the CROSSalign method [89], which predicts structural similarities between two RNAs according to the hybridization probabilities, neighboring nucleotides, and cellular environment [90]. According to the criterion of Delli-Ponti [89], a CROSSalign distance lower than 0.08 and *p*-value < 10^−5^ determines structural similarity. However, in this research, no transcripts met these criteria. Therefore, we studied the five transcripts with a normalized structural distance < 0.1 and a *p*-value < 0.05 (Figure 4) to discover possible structural similarities among lncRNAs, since a *p*-value > 0.10 indicates a non-significant score [89].

The lncRNAs that met these characteristics included Solyc04r022210.1 (*S. lycopersicum*), BNAP_LNC003827.4 (*B. napus*), BRAP_LNC003139.3 (*B. rapa*), MTRU_LNC004005.50 (*M. truncatula*), ACHI_LNC008871.8 (*A. chinensis*), and S03 (*L. mutabilis*). As shown in Figure 4A, the lncRNA S03 from *L. mutabilis* shares a degree of structural conservation with Solyc04r022210.1 in *S. lycopersicum*, with a normalized structural distance value of the secondary structure profiles of 0.095, obtained in the region 115–790 bp and that shows a correlation of 0.91 with a significance of 0.033. The matrix also allowed us to establish the similarity between BNAP_LNC003827.4 in *B. napus* and BRAP_LNC003139.3 in *B. rapa*. These lncRNAs had a structural distance value of 0.047 between 267 and 997 nucleotides, with a correlation of 0.94 and a significance *p*-value = 0.00013. Lastly, the transcripts BRAP_LNC003139.3 from *B. rapa* and ACHI_LNC008871.8 from *A. chinensis* showed a structural distance value of 0.093 in the region 265–1021, with a correlation of 0.92 and a *p*-value = 0.016.

The structural analysis was completed with MEME [91], which was used to compare the conserved motifs among lncRNAs. This software predicted the presence of five motifs in the transcripts from the six species (Figure 4B). However, motif five was not found in *L. mutabilis* lncRNA S03. The similarities among secondary structures were evident at the level of a multibranched loop circled in Figure 4C. This global topological pattern, the sum of the physical or logical arrangements of nodes and connections inside a network, is more important than the individual base pairings [92] and indicates an interactive role with different elements at a molecular level [93].

### 2.6. Expression and Validation of lncRNA S03

lncRNAs participate in different metabolic processes, such as gene silencing, maturation, reproduction, differentiation, and stress tolerance, by interacting with multiple elements at a molecular level [94]. Drought is the most detrimental stress to agricultural production worldwide [95], causing losses of almost 20% of potential yields [96]. Extreme droughts have recently been reported to affect the Peruvian Andes [97]. Therefore, only lncRNA S03 was validated under drought stress conditions (Figure 5A), because no primers could be designed for S05, S07, S08, S09, or S10 and the amplification failed for the other lncRNAs (Appendix A). Coincidentally, the lncRNA S03 validated in this study is the only one that achieved structural stability according to the criterium of MFE value < −80 kcal/mol [65].

In *L. mutabilis* under progressive drought, we found a high expression of S03 in contrast to the control (Figure 5B). Since this lncRNA was being expressed in a drought-tolerant accession and included structural elements capable of interacting with different molecular targets (Figure 3B), we propose that S03 is important in response to drought stress in *L. mutabilis*. Moreover, the SECIS element (Figure 3) contained in mRNAs is required for the synthesis of selenoproteins, which are required for Se homeostasis [98]. Thus, S03 may have a critical role in Se homeostasis under drought. Indeed, Ahmad et al. [99] reported that Se might regulate water status and increase biomass through the activation of glutathione peroxidase in water-stressed plants.

Jampala et al. [100] mention that several lncRNAs are associated with drought tolerance in other plant species. For instance, in *A. thaliana*, the lncRNA DRIR (drought induced lncRNA) is overexpressed under dehydration and ABA treatment, acting as a positive regulator of stress tolerance, regulating ABA response, and inducing salinity and drought tolerance [9]. These findings are similar to the report of Chen et al. [101] regarding the transgenic lines of *G. max*, where lncRNA77580 improved the tolerance to drought and yield under stress conditions. Likewise, Li et al. [11] found that the lincRNA340 represses miR169 in *M. esculenta*, inducing the expression of nuclear factor Y (NF-Y) to increase drought tolerance by acting as a mimetic target of miRNAs and transcripts associated with transduction pathways in the hormone signaling, metabolite biosynthesis, and sucrose metabolism [11].

## 3. Materials and Methods

### 3.1. Transcriptomic Data and Quality Control

Two RNA-Seq experiments were performed in the accessions P3 and P11 of *L. mutabilis* given to the Universidad Nacional Agraria La Molina under a material transfer agreement (ATMG 001–2015). The first experiment analyzed the transcriptome of flower buds in two developmental stages [26]; the second studied the vegetative tissue response to drought stress [27].

Twelve RNA-seq libraries were prepared by MACROGEN Inc., Seoul, Korea. Total RNA sequencing through Illumina RNA-seq was performed with the sequencer HiSeq 2500 to obtain paired-end reads. Each experiment consisted of two treatments with three biological replicates. Low-quality reads with Phred score < 30, short reads (<20 bp), empty nucleotides (N at the end of reads), and adapter sequences were trimmed using CutAdapt software to obtain high-quality reads. Quality control before and after trimming was performed using FastQC according to Legget et al. [102]. Raw data comprised 598,479,350 initial reads that produced 581,249,244 high-quality reads. Finally, these high-quality reads from both studies were assembled de novo using TRINITY to obtain the total transcripts, producing contigs with a GC percentage of 38.62%. The contigs had an N50 = 1879, with an average of 1184.59 bp. TRINITY assembly produced 172,370 transcripts and 83,145 total genes. The HiSeq raw data Macrogen report is summarized in Appendix A.

### 3.2. Putative lncRNA Identification

Transcripts were filtrated following the methodology of Khemka et al. [15]. Therefore, the transcripts were filtrated to obtain the putative lncRNAs. The workflow is shown in Figure 6. Transcripts with a length ≤200 bp and open reading frames (ORF) with a length >300 bp were discarded as potentially coding transcripts in *A. thaliana*, *G. max*, *Medicago truncatula*, and *L. angustifolius* [103] with SwissProt in Pfam.

Then, the coding potential of the transcripts was evaluated using four different algorithms: CPC [30], CPC2 [104], CNIT [33], and lncFinder [39]. The transcripts with values of CPC < 0; CPC2 < 0.5; CNIT < 0; lncFinder = 0 were considered lncRNAs. The selection of putative lncRNAs was performed by the coincidence of at least two algorithms, following the criterion of Aslam et al. [18].

### 3.3. Putative lncRNA Annotation

The functional annotation of lncRNAs was performed against a database including PLncDB v2.2 [105], GreeNC [106], CANTATAdb 2.0 [107], RNAcentral [108], and Ensembl 2.0 [109]. According to the methodology of Hezroni et al. [43], the multiple sequence alignment was performed with the BLASTN from the suite BLAST+ 2.9.0, with the modified parameters “-task blastn -word_size 8 -strand plus”. Following the modified Khoei et al. [44] criteria, the transcripts with alignments with E-value < 10^−5^, identity > 70%, and alignment length 100 bp were considered homologs. As a selection criterion, the *L. mutabilis* lncRNAs with homologs with 40 other species or more were taken as conserved lncRNAs.

### 3.4. Subcellular Location

The subcellular location databases RNAlocate [49], iLoc-LncRNA (2.0) [50], DeepLncLoc [51], and lncLocator 2.0 [52] were used in the default settings to identify the subcellular location of each transcript in *L. mutabilis*, which were taken as a consensus according to Yang et al. [110].

### 3.5. Thermodynamic Characterization of lncRNAs

According to Bugnon et al. [70], the characterization of lncRNAs was performed by searching the structures with the minimum free energy (MFE) values using the average values of the algorithms RNAfold [61], LinearFold [62], LinearPartition [63], and RNAshapes [64]. MFE values < −80 kcal/mol were considered stable, following Mohammadin et al.’s [65] criterion.

### 3.6. Structural Conservation of Transcripts

CROSSalign [89] was used to compare the secondary structure of the stable lncRNAs (MFE < −80 kcal/mol) against their homologs found in the putative lncRNA annotation. A matrix was built to evaluate the similarity based on the combination of the computational recognition algorithm of secondary structure (CROSS) to predict the secondary structure profile of the RNA with a resolution of single nucleotide and the dynamic time warping (DTW) to align profiles of different lengths. The evaluation of the significance was performed according to the criterion of Delli-Ponti [89], in which a structural distance value lower than 0.08 and *p*-value < 10^−5^ indicated significant structural similarity and *p*-values of 0.10 or higher resulted in non-significant scoring. The RNAFold Ct format of each lncRNA was plotted with RiboSketch to search for similarities among secondary structures. Lastly, the MEME (v. 5.5.3) Motif Discovery software [91] was used to predict new conserved motifs.

### 3.7. Validation of Differential Expression Patterns by RT-PCR

To validate the expression patterns, the RT-PCR was performed with the plant material from vegetative tissues in normal and drought conditions by duplicate, according to Hidalgo [27]. The total RNA extraction and the genomic DNA removal were performed according to the kit instructions for Direct-zol™ RNAMiniprepPlus (ZYMO RESEARCH, Irvine, CA, USA). The total RNA samples were evaluated with the nanophotometer IMPLEN NP80 and the quality was assessed with an electrophoresis analysis. The DNAse treatment was performed with the kit RQN from Promega, according to the instructions. cDNA synthesis was performed according to the kit RevertAid First Strand cDNA Synthesis Kit instructions using random hexamer primers. The specific primers for lncRNAs were designed with Primer3web [111], version 4.1.0, used for the amplification with the BioRad T100 thermal cycler with three biological replicates and three technical ones. Differential expression analysis was evaluated using the reference genes reported by Linares [112]. These genes were evaluated under drought conditions and OEP24 was selected as an internal control for normalization. The sequences for the primers used for the amplification of lncRNAs and OEP24 in *L. mutabilis* are shown in Table 2.

## 4. Conclusions

This research provides valuable information on novel lncRNAs in *L. mutabilis* by re-porting 590 putative lncRNAs according to two algorithms (CPC, CPC2, CNIT, or lncFinder). The application of the homology criterion of Hezroni et al. [37] enabled the identification of S03, a lncRNA with an MFE value = 105.67kcal/mol, that included a stem and two multibranch loops with a SECIS-like element in its structure. This element is required for selenoproteins, which are involved in Se homeostasis and might play a critical role in drought response by regulating water status and increasing biomass through the activation of antioxidant mechanisms. The evaluation of the structural conservation by CROSSalign showed partial similarity between S03 in *L. mutabilis* and Solyc04r022210.1 in *S. lycopersicum*. The lncRNA S03 was upregulated in the drought-tolerant accession P03 of *L. mutabilis*. Finally, these results highlight the importance of lncRNAs in tarwi improvement under drought conditions. However, further experiments on specific lncRNAs are still needed to elucidate their exact function.

## Figures and Tables

**Figure 1 ncrna-09-00048-f001:**
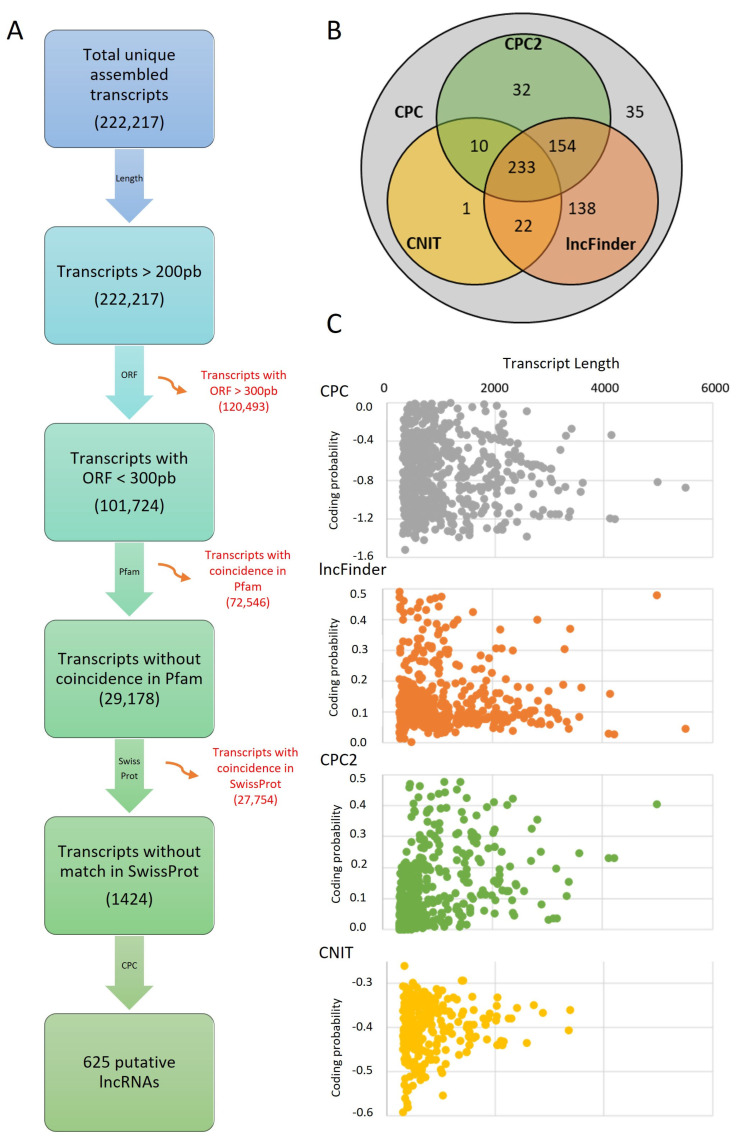
LncRNA prediction in *L. mutabilis*. (**A**) lncRNA identification, characterization, and validation in *L. mutabilis*. The filter applied is described in the arrows. The colored charts detail the information included and the red arrows describe the excluded data. (**B**) Venn diagram for the CPC, lncFinder, CPC2, and CNIT algorithms used in lncRNA prediction. (**C**) Coding probability vs. transcript length according to the CPC, lncFinder, CPC2, and CNIT algorithms.

**Figure 2 ncrna-09-00048-f002:**
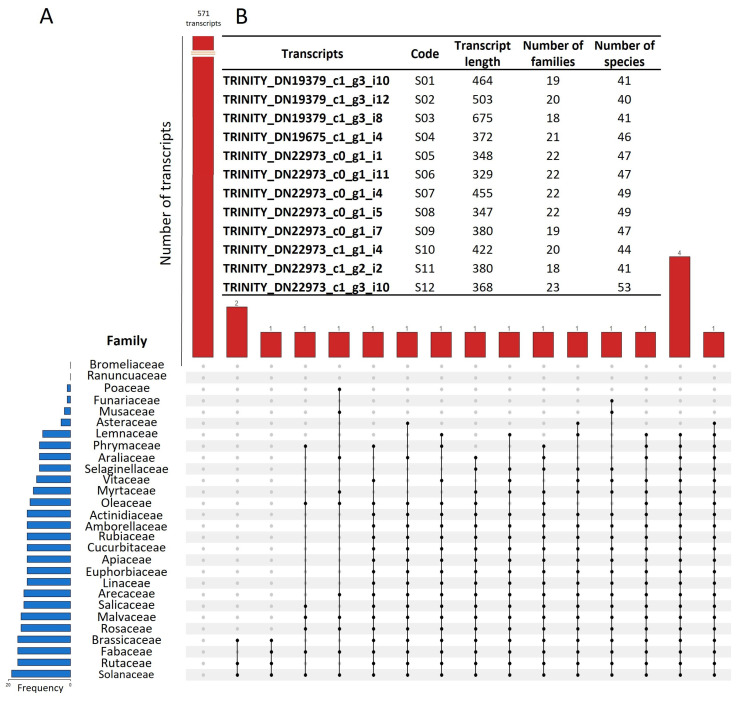
*L. mutabilis* lncRNA annotation against the PLncDB v2.2 database. (**A**) Upset plot showing the abundance of transcripts by botanical families. (**B**) *L. mutabilis* predicted lncRNAs with the highest homologs within the database.

**Figure 3 ncrna-09-00048-f003:**
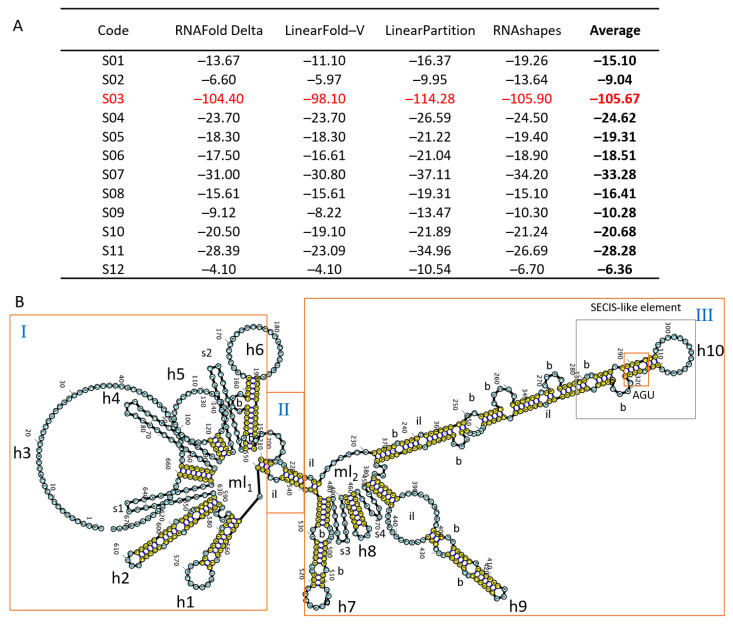
Thermodynamic and structural characterization of *L. mutabilis* lncRNAs. (**A**) Minimum free energy for the transcripts of tarwi with the highest number of homologs according to different algorithms. The average MFE value for each transcript is reported in bold. The transcript with an MFE indicating stability is colored in red. (**B**) Secondary structure of S03 according to RNAFold (MFE <−80 kcal/mol; Mohammadin et al. 2015). (**I**) A multibranched loop with eight branches. (**II**) Stem with internal loop. (**III**) Multiloop with six branches, including the SECIS-like element with the AGU motif; h—hairpin; ml—multiloop; hl—hairpin-loop; el—external loop, il—internal loop; b—bulge and s—stem.

**Figure 4 ncrna-09-00048-f004:**
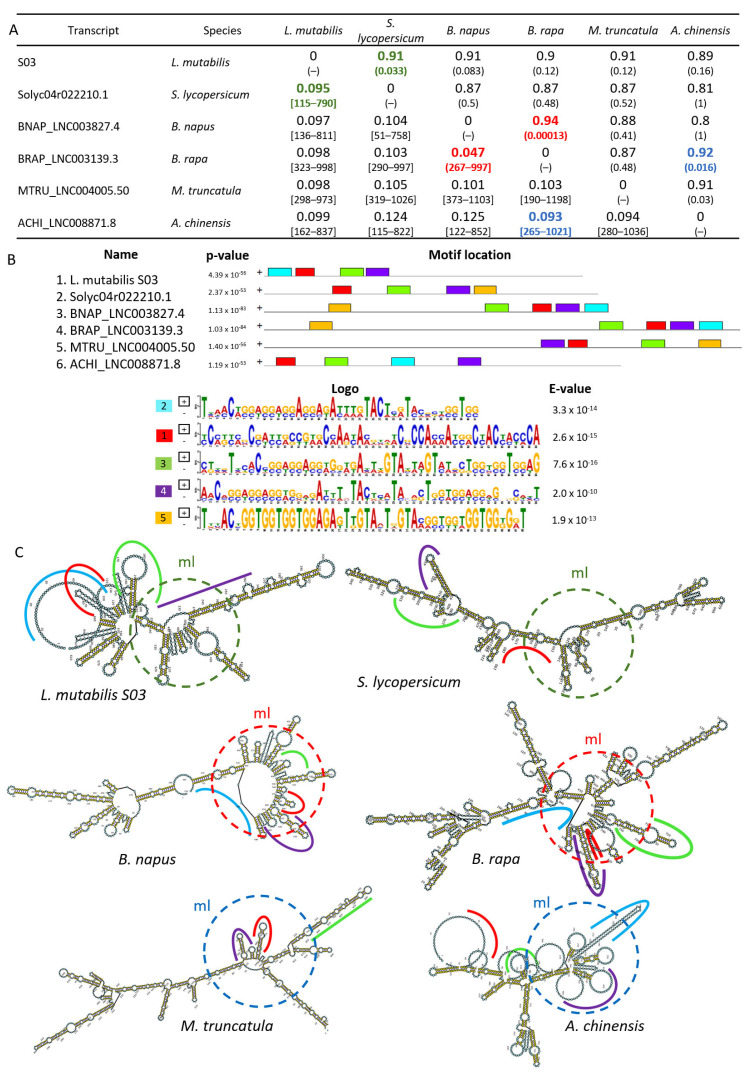
Structural conservation of *L. mutabilis* S03. (**A**) CROSSallign distances between *L. mutabilis* S03 and its homologs. Under the diagonal are data and region of the normalized structural distance (square brackets) of the secondary structure profiles among transcript RNA pairs. Above the diagonal are the values of correlation and significance (parenthesis). The conserved structures with significant *p* values are reported in bold and colored according to the similarities among secondary structures. (**B**) Motif logos from MEME motif discovery. Corresponding e-values from MEME are listed. (**C**) Predicted secondary structures of lncRNAs modeled in RNAFold. Colored lines show aligned regions in the long profile. Dotted circles with the same color show a global topological pattern.

**Figure 5 ncrna-09-00048-f005:**
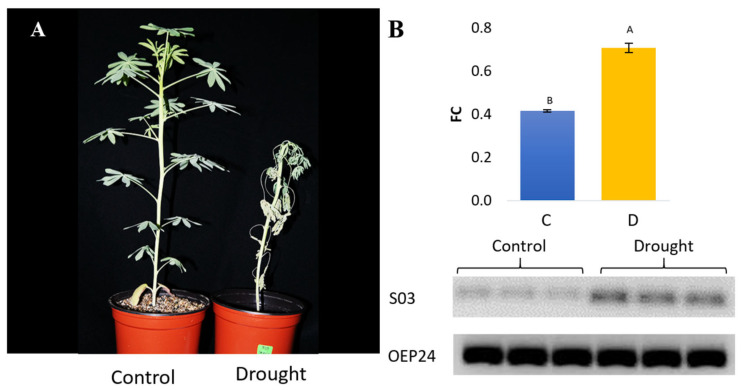
RT-PCR Validation of *L. mutabilis* S03 under drought stress. (**A**) *L. mutabilis* at fifth-leaf phenological stage under ten days of progressive drought stress. (**B**) Folding change (FC) of S03 under drought and normal conditions. The gene OEP24 (Linares 2022) was used as a normalization reference. Capped lines express the standard error. Different letters indicate significant differences according to Tukey HSD (alpha < 0.05).

**Figure 6 ncrna-09-00048-f006:**
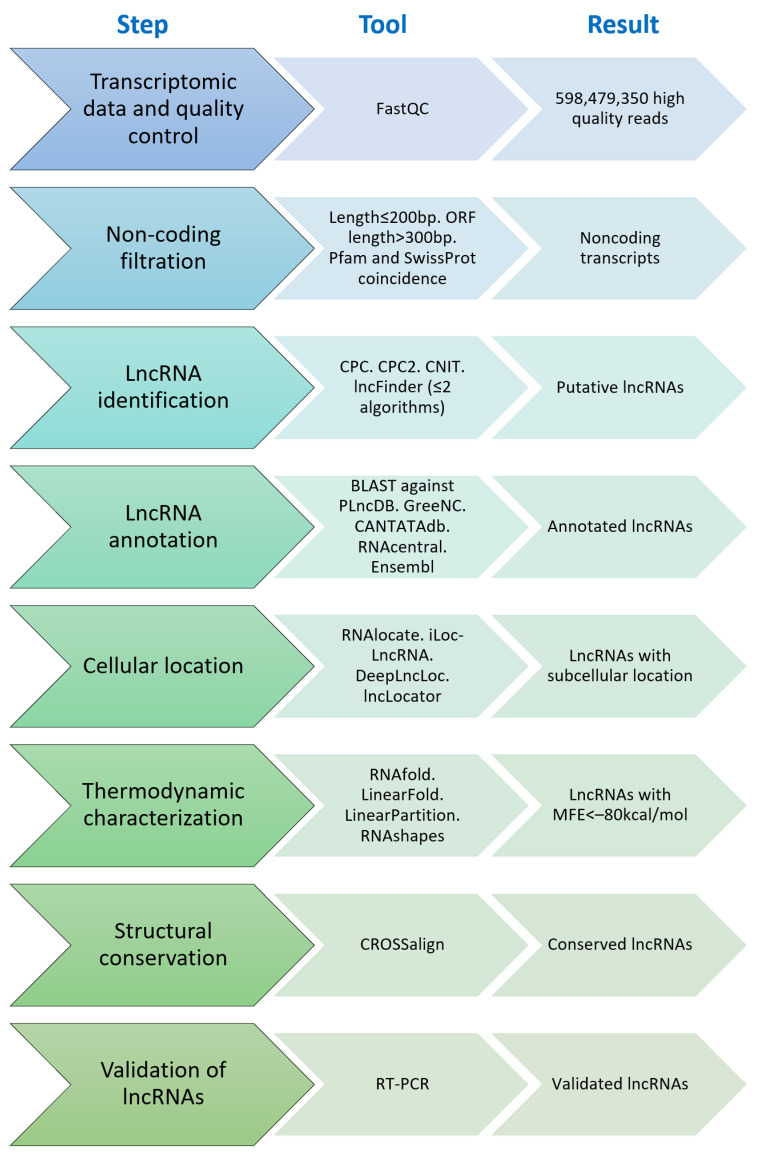
Workflow for lncRNA identification, characterization, and validation in *L. mutabilis*.

**Table 1 ncrna-09-00048-t001:** Subcellular location of *L. mutabilis* lncRNAs according to four algorithms. The numbers in brackets indicate the probability of the location of the transcript according to the respective algorithm. Consensus location and sublocation are reported in bold.

Transcript ID	Code	RNALOCATE	iLoc-LncRNA 2.0	DeepLncLoc	lncLocator 2.0	Predicted Location	Predicted Sublocation
TRINITY_DN19379_c1_g3_i10	S01	Ribosome (0.99)	Ribosome (0.99)	Cytoplasm (0.56)	Nucleus (−1.04)	**Cytoplasm**	**Ribosome**
TRINITY_DN19379_c1_g3_i12	S02	Ribosome (0.58)	Ribosome (0.58)	Cytoplasm (0.49)	Nucleus (−1.11)	**Cytoplasm**	**Ribosome**
TRINITY_DN19379_c1_g3_i8	S03	Cytoplasm. Cytosol (0.70)	Cytoplasm. Cytosol (0.70)	Cytoplasm (0.62)	Nucleus (−0.56)	**Cytoplasm**	**Cytosol**
TRINITY_DN19675_c1_g1_i4	S04	Cytoplasm. Cytosol (0.99)	Cytoplasm. Cytosol (0.99)	Cytoplasm (0.67)	Nucleus (−1.31)	**Cytoplasm**	**Cytosol**
TRINITY_DN22973_c0_g1_i1	S05	Ribosome (0.53)	Ribosome (0.53)	Cytoplasm (0.67)	Nucleus (−0.62)	**Cytoplasm**	**Ribosome**
TRINITY_DN22973_c0_g1_i11	S06	Ribosome (0.72)	Ribosome (0.72)	Cytoplasm (0.76)	Nucleus (−1.20)	**Cytoplasm**	**Ribosome**
TRINITY_DN22973_c0_g1_i4	S07	Ribosome (0.82)	Ribosome (0.82)	Cytoplasm (0.63)	Nucleus (−1.18)	**Cytoplasm**	**Ribosome**
TRINITY_DN22973_c0_g1_i5	S08	Exosome (0.83)	Exosome (0.83)	Cytoplasm (0.64)	Nucleus (−1.16)	**Exosome**	**Exosome**
TRINITY_DN22973_c0_g1_i7	S09	Exosome (0.67)	Exosome (0.67)	Cytoplasm (0.65)	Nucleus (−1.14)	**Exosome**	**Exosome**
TRINITY_DN22973_c1_g1_i4	S10	Exosome (0.77)	Exosome (0.77)	Cytoplasm (0.48)	Nucleus (1.32)	**Exosome**	**Exosome**
TRINITY_DN22973_c1_g2_i2	S11	Exosome (0.59)	Exosome (0.59)	Cytoplasm (0.71)	Nucleus (−1.05)	**Exosome**	**Exosome**
TRINITY_DN22973_c1_g3_i10	S12	Ribosome (0.83)	Ribosome (0.83)	Cytoplasm (0.68)	Nucleus (−1.04)	**Cytoplasm**	**Ribosome**

**Table 2 ncrna-09-00048-t002:** RT-PCR primer list for lncRNAs validation in *L. mutabilis*.

Gene Name (*L. mutabilis* Transcript ID)	Sequence (5′–3′)	Product Size (bp)	Annealing Temperature (°C)
OEP24k (TRINITY_DN20247_c5_g3_i3)	Forward: GTCTAAGAACTCGTGGGACTTTG Reverse: CATGTGGTCTCTGCACTAAGTTT	201	60.9
S03 (TRINITY_DN19379_c1_g3_i8)	Forward: TCACTACTAGGCTGAGCAACC Reverse: TGTTCCCTGCTTCTTCTTGTG	111	60.4

## Data Availability

The datasets generated during the current study are available at GenBank OR244634-OR245258.

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
