# Peer review of "Analysis of lncRNAs in Lupinus mutabilis (Tarwi) and Their Potential Role in Drought Response"

_ncrna, 2023, doi:10.3390/ncrna9050048_

Round 1
Reviewer 1 Report
Significance:
A manuscript by Manuel Hidalgo entitled “Analysis of lncRNAs in Lupinus mutabilis (tarwi) and their potential role in drought response “is a structured study aimed to characterize the drought response induced lncRNA in Lupinus mutabilis.
Major comments:
1. The authors used the detailed characterization of lncRNAs from various RNA-Seq data but in the MS, they only tested on lncRNA S03 expression induced in drought.
2. Starting with two different studies with highly variable numbers of RNA (also lncRNAs) being identified reduces the impact of the study.
3. Authors can also check lncRNAs with similar secondary structures and RNA fold as of S03 in drought conditions.
Minor comments:
- The introduction can be modified by not only giving examples but highlighting their importance in the present study.
- In Conclusion, please discuss the possible mechanism of upregulation of S03 in response to drought.
There are minor grammatical corrections that can be improved.
Author Response
A manuscript by Manuel Hidalgo entitled “Analysis of lncRNAs in Lupinus mutabilis (tarwi) and their potential role in drought response “is a structured study aimed to characterize the drought response induced lncRNA in Lupinus mutabilis.
Major comments:
- The authors used the detailed characterization of lncRNAs from various RNA-Seq data but in the MS, they only tested on lncRNA S03 expression induced in drought.
Answer: In line [351-357], It was changed as followed: “Therefore, only the lncRNA S03 was validated under drought stress conditions (Figure 5A), because no primers could be designed for S05, S07, S08, S09 and S10; and the amplification failed for the other lncRNAs (Supplementary Table S3). Coincidentally, the lncRNA S03 validated in this study, is the only one that achieves structural stability according to the criterium of MFE value < −80 kcal/mol (Mohammadin et al. [59]).”.
Line [466] We are also stating that the validation was performed by duplicated. Moreover, the Supplementary Table S3 is added.
- Starting with two different studies with highly variable numbers of RNA (also lncRNAs) being identified reduces the impact of the study.
Answer: Line [395-407] The detailed description has been added to the methodology section. Moreover, the Supplementary Table S4 is added.
- Authors can also check lncRNAs with similar secondary structures and RNA fold as of S03 in drought conditions.
Answer: The search for lncRNAs, including the SECIS element in drought studies, has been done to compare them with S03. Thus, structural conservation has been evaluated with CROSSALLIGN, only finding similarity with the lncRNA of S. lycopersicum (Fig. 4A; supplementary Table S2). However, the SECIS element was not found in tomato. Therefore, Meme was used to discover new motifs (Fig 4B). Finally, we discussed the global topological pattern in lines [330-333].
Minor comments:
The introduction can be modified by not only giving examples but highlighting their importance in the present study.
Answer: Line [61-65] The introduction has been modified including the importance of lncRNA in drought response.
In Conclusion, please discuss the possible mechanism of upregulation of S03 in response to drought.
Answer: This mechanism is discussed in the Validation section (lines 362-366) and restated in the conclusion (lines 486-489).
Reviewer 2 Report
I hope this letter finds you well. I am writing to express my review and evaluation of the article titled "Analysis of lncRNAs in Lupinus mutabilis (tarwi) and their potential role in drought response," submitted to non-coding RNA. As a peer reviewer, I have thoroughly assessed the manuscript and would like to emphasize the commendable aspects of the work.
Firstly, I must commend the authors on the quality of their research. The article is exceptionally well-written, with a coherent and continuous flow of information. The clarity of language and presentation greatly enhances the readability of the manuscript, making it accessible to both experts and non-specialists interested in the subject matter. The utilization of up-to-date and relevant sources is another commendable aspect of this research. The authors have demonstrated a keen understanding of the field, and their incorporation of pertinent literature adds credibility and value to the study.The research methodology employed in this study is appropriate and well-suited to address the stated research objectives. The authors have employed a rigorous approach to analyze lncRNAs in Lupinus mutabilis.
However, there are some points to consider and answer:
1. Sample Size: The article only examines two samples, which reduces the statistical power of the analysis and makes it challenging to identify genuine patterns and effects in the data. A larger sample size is essential for stronger empirical validation and to minimize sampling bias.
2. Experimental Validation: While experimental validation of one predicted lncRNA using RT-PCR under drought conditions is valuable, it is recommended to validate at least 5 to 10 lncRNAs with the highest probability of playing a role in the drought response. Confirming multiple lncRNAs through molecular experiments strengthens the credibility of the findings. Additionally, validating more than one lncRNA allows for the identification of common patterns among them, leading to a better understanding of their role in response to drought.
It is important to address these points in the article to enhance the overall quality of the research. Providing a more comprehensive and robust experimental validation will improve the reliability and impact of the study's findings.
Author Response
I hope this letter finds you well. I am writing to express my review and evaluation of the article titled "Analysis of lncRNAs in Lupinus mutabilis (tarwi) and their potential role in drought response," submitted to non-coding RNA. As a peer reviewer, I have thoroughly assessed the manuscript and would like to emphasize the commendable aspects of the work.
Firstly, I must commend the authors on the quality of their research. The article is exceptionally well-written, with a coherent and continuous flow of information. The clarity of language and presentation greatly enhances the readability of the manuscript, making it accessible to both experts and non-specialists interested in the subject matter. The utilization of up-to-date and relevant sources is another commendable aspect of this research. The authors have demonstrated a keen understanding of the field, and their incorporation of pertinent literature adds credibility and value to the study.The research methodology employed in this study is appropriate and well-suited to address the stated research objectives. The authors have employed a rigorous approach to analyze lncRNAs in Lupinus mutabilis.
However, there are some points to consider and answer:
- Sample Size: The article only examines two samples, which reduces the statistical power of the analysis and makes it challenging to identify genuine patterns and effects in the data. A larger sample size is essential for stronger empirical validation and to minimize sampling bias.
Answer: Line [395-407] The detailed description has been added to the methodology section. Moreover, the Supplementary Table S4 is added.
- Experimental Validation: While experimental validation of one predicted lncRNA using RT-PCR under drought conditions is valuable, it is recommended to validate at least 5 to 10 lncRNAs with the highest probability of playing a role in the drought response. Confirming multiple lncRNAs through molecular experiments strengthens the credibility of the findings. Additionally, validating more than one lncRNA allows for the identification of common patterns among them, leading to a better understanding of their role in response to drought.
Answer: In line [351-357], it was changed as followed: “Therefore, only the lncRNA S03 was validated under drought stress conditions (Figure 5A), because no primers could be designed for S05, S07, S08, S09 and S10; and the amplification failed for the other lncRNAs (Supplementary Table S3). Coincidentally, the lncRNA S03 validated in this study, is the only one that achieves structural stability according to the criterium of MFE value < −80 kcal/mol (Mohammadin et al. [59]).”.
Line [466] We are also stating that the validation was performed by duplicated. Moreover, the Supplementary Table S3 is added.
It is important to address these points in the article to enhance the overall quality of the research. Providing a more comprehensive and robust experimental validation will improve the reliability and impact of the study's findings.
Answer: Thanks for the comments and the MS was corrected.
Reviewer 3 Report
At the start of the abstract, please add the problem statement, like the urgency of the current study.
Line 12, do not start a sentence with numbers.
Kindly use some diverse keywords.
Line 28-35, kindly add a recent study about miRNA/lncRNAs such as doi: 10.1016/j.plaphy.2023.107857
In the introduction, there is no information about drought stress. Please add a comprehensive paragraph about drought stress (doi: /10.1002/tpg2.20279).
In the results, please do not simply present what was found. Authors also need to focus on what their data reflect.
Author Response
At the start of the abstract, please add the problem statement, like the urgency of the current study.
Answer: Line [11] The problem has been stated.
Line 12, do not start a sentence with numbers.
Answer: Line [17] done.
Kindly use some diverse keywords.
Answer:Done.
Line 28-35, kindly add a recent study about miRNA/lncRNAs such as doi: 10.1016/j.plaphy.2023.107857
Answer: Lines [33] and [63-65]: done.
In the introduction, there is no information about drought stress. Please add a comprehensive paragraph about drought stress (doi: /10.1002/tpg2.20279).
Answer: Lines [66-73]: A comprehensive paragraph about drought stress has been added in the introduction
In the results, please do not simply present what was found. Authors also need to focus on what their data reflect.
Answer: Lines [362-366] and [487-490]: The putative mechanism has been added.
Round 2
Reviewer 1 Report
Thanks for answering our concerns.
Minor improvement
Reviewer 3 Report
The revised version can be accepted.
Minor editing of English language required